# Light Scattering and Turbidimetry Techniques for the Characterization of Nanoparticles and Nanostructured Networks

**DOI:** 10.3390/nano12132214

**Published:** 2022-06-28

**Authors:** Pietro Anzini, Daniele Redoglio, Mattia Rocco, Norberto Masciocchi, Fabio Ferri

**Affiliations:** 1Dipartimento di Scienza e Alta Tecnologia and To.Sca.Lab, Università degli Studi dell’Insubria, Via Valleggio 11, 22100 Como, Italy; pietro.anzini@uninsubria.it (P.A.); daniele.redoglio@gmail.com (D.R.); norberto.masciocchi@uninsubria.it (N.M.); 2Proteomica e Spettrometria di Massa, IRCCS Ospedale Policlinico San Martino, Largo R. Benzi 10, 16132 Genova, Italy; mattia.rocco@quipo.it

**Keywords:** dynamic light scattering, static light scattering, turbidimetry, colloidal aggregation, fractal morphology

## Abstract

Light scattering and turbidimetry techniques are classical tools for characterizing the dynamics and structure of single nanoparticles or nanostructured networks. They work by analyzing, as a function of time (Dynamic Light Scattering, DLS) or angles (Static Light Scattering, SLS), the light scattered by a sample, or measuring, as a function of the wavelength, the intensity scattered over the entire solid angle when the sample is illuminated with white light (Multi Wavelength Turbidimetry, MWT). Light scattering methods probe different length scales, in the ranges of ~5–500 nm (DLS), or ~0.1–5 μm (Wide Angle SLS), or ~1–100 μm (Low Angle SLS), and some of them can be operated in a time-resolved mode, with the possibility of characterizing not only stationary, but also aggregating, polymerizing, or self-assembling samples. Thus, the combined use of these techniques represents a powerful approach for studying systems characterized by very different length scales. In this work, we will review some typical applications of these methods, ranging from the field of colloidal fractal aggregation to the polymerization of biologic networks made of randomly entangled nanosized fibers. We will also discuss the opportunity of combining together different scattering techniques, emphasizing the advantages of a global analysis with respect to single-methods data processing.

## 1. Introduction

Light Scattering (LS) techniques refer to a family of experimental optical methods based on the phenomenon of scattering, which occurs whenever a beam of light impinging onto an optically inhomogeneous sample produces radiation that is diffused away from the incident direction. Since the advent of the laser in the 1960s, LS has been increasingly used for studying a large variety of so-called soft-matter systems, with applications both in fundamental and applied science. Examples include the fields of colloidal aggregation, polymer blends, gel formation, and, in general, the chemical physics of complex fluids and critical phenomena [1,2,3,4,5].

LS techniques have the great advantage of being non-invasive and providing information almost in real time. They are also highly reliable from a statistical point of view because they are applied to samples made of a high number of particles. Among the various LS techniques, the most popular ones are Dynamic Light Scattering (DLS), Static Light Scattering (SLS), and Multi-Wavelength Turbidimetry (MWT or, simply, turbidimetry).

Static Light Scattering (SLS) is based on the measure of the time-averaged angular distribution of light intensity elastically scattered by a sample. SLS provides information on molecular weight, average size (gyration radius), and morphological structure of the scatterers dispersed in a suspension. SLS techniques can also be operated in a time-resolved mode, with the possibility of characterizing not only stationary but also aggregating, polymerizing, or self-assembling samples. In principle, the angular range accessible to SLS can be very wide, from ~0.1° (or even smaller) to ~180°, but there is no single instrument capable of covering such a wide range, and quite different experimental approaches must be used: in the ~0.1–10° range (Low Angle SLS, LA-SLS) bi-dimensional detectors, such as CCD or array of photodiodes are exploited, whereas at larger angles, in the ~10–170° range (Wide Angle SLS, WA-SLS) highly sensitive point-like detectors such as photomultipliers or avalanche photodiodes must be used. Depending on the angular range, the length scales (or typical particle sizes) probed by SLS vary between ~0.1–5 μm (WA-SLS) or ~1–100 μm (LA-SLS).

Dynamic Light Scattering (DLS) measures the time correlation function of the fluctuations of the intensity scattered by the investigated sample at a given angle and provides information on the decay or relaxation time (or times) that characterizes its underlying dynamics. Examples of applications of DLS are countless, the most prominent ones probably being particle sizing of nanosized particles via the measurement of the translational diffusion coefficient associated with their Brownian motion. DLS is routinely utilized in many laboratories worldwide, with applications ranging from industrial production control to the fundamental study of interacting particle systems. Typical diameters recoverable with DLS are in the ~5–500 nm range.

Multi-Wavelength Turbidimetry (MWT) consists of the measurement of the sample extinction coefficient as a function of the wavelength of the incident radiation, typically in a spectral range that covers the UV-VIS NIR region (300–1000 nm). MWT is not properly a scattering technique because it does not directly measure the scattered light at various angles but only the overall power transmitted by the sample. In general, the transmitted power is attenuated (with respect to the incident one) because of absorption and scattering, but for non-absorbing samples or in the spectral regions where there is no absorption, the attenuation is due only to scattering and, therefore, the MWT technique can be considered a truly (integrated) scattering technique, probing length scales typical of WA-SLS.

In this mini-review, we will recall the fundamentals of these three techniques and report some examples of the experimental setups necessary to implement them. Then, we will review a few typical applications of these methods, ranging from the field of colloidal fractal aggregation to the polymerization of biologic networks made of randomly entangled (nanosized) fibers. Finally, we will discuss the advantages of coupling together some of these techniques, emphasizing the benefits of a global analysis with respect to single-methods of data processing. As this review deals with light scattering techniques, we have not included other well-established techniques such as Small-Angle X-ray or Neutron Scattering (SAXS and SANS), which clearly can provide additional important information and extend the type and range of sample analysis.

## 2. Theoretical Background

In the theory of Light Scattering (LS), the light scattered away from the incident beam is due to the presence of local fluctuations of the dielectric constant of the medium ε over the entire scattering volume V. In many cases, such as for colloidal, macromolecular, or gel systems, these fluctuations are due to the presence of particulate scatterers that have a refraction index different from that of the medium.

Let us suppose to have an optically inhomogeneous, non-absorbing, non-magnetic, non-conductive, and isotropic medium characterized by a time-dependent dielectric constant $\varepsilon \left(r,t\right)$. Let the medium be illuminated with a linearly polarized incident monochromatic electric field of amplitude $E_{0}$, oscillating at a frequency $\omega_{0}$ with a vacuum wavelength $\lambda_{0}=2\pi c/\omega_{0}$. If the fluctuations $\delta \varepsilon \left(r,t\right)=\varepsilon \left(r,t\right)-\langle \varepsilon \rangle$ are small with respect to the average dielectric constant of the medium 〈ε〉, i.e., when $\delta \varepsilon \left(r,t\right)\ll \langle \varepsilon \rangle$, the Born approximation applies and at a very large distance $R\gg r\left(t\right)$ (far-field limit), and the amplitude of the scattered field reads [2].
(1)$$E\left(q,t\right)=\frac{\pi }{\lambda_{0}^{2}}E_{0}e^{-i\omega_{0}t}\frac{e^{ikR}}{R}\sin\left(\phi \right)\underset{V}{\int }\delta \varepsilon \left(r,t\right)e^{iq\cdot r}dr$$
where q is the scattering wavevector defined as $q=k-k_{0}$, and k and $k_{0}$ are the scattered and the incident wavevectors. The angle between k and $k_{0}$ defines the scattering angle θ. In Equation (1) $e^{ikR}/R$ is a spherical wave term, ϕ is the angle between k and the polarization direction of the incident electric field, V is the scattering volume and $e^{iq\cdot r}$ is a phase term that describes the interference between the electric fields scattered as spherical waves by all the infinitesimal sub-volumes dr.

Note that when the fluctuations are frozen [$\delta \varepsilon \left(r,t\right)=\delta \varepsilon \left(r\right)$], the integral does not depend on time, and the scattered field oscillates at the same frequency of the incident radiation (Elastic or Static Light Scattering, ELS, or SLS). Consequently, $\left|k\right|=\left|k_{0}\right|$ and the magnitude of q is related to the scattering angle θ by the relation $q=\left(4\pi /\lambda_{0}\right)n_{0}\sin\theta /2$. Conversely, when $\delta \varepsilon \left(r,t\right)$ moves at velocity $\left|v\right|\ll c$, the scattered field undergoes a Doppler shift of the order of $∆\omega \sim \left|v\right|/c\ll \omega_{0}$ and, therefore, oscillates at almost the same frequency as the incident field (Quasi Elastic Light Scattering or Dynamic Light Scattering, QELS, or DLS). Finally, it is worth recalling that Einstein was the first one, in late 1910 [6], to describe the scattering as the result of the local fluctuations of the medium dielectric constant, exactly as reported in Equation (1).

Discrete scatterers

Let us now suppose that the medium is a suspension of particles with an index of refraction different from that of the solvent. Let N be the number of particles in the scattering volume V and indicate with $R_{k}\left(t\right)$ $\left(k=1,\ldots ,\;N\right)$ the positions of their centers of mass at time t, whereas $r_{k}$ indicates the position of each particle element with respect to the corresponding center of mass. Under the assumption that the scattering from the solvent is negligible with respect to that of the particles (see Appendix A), Equation (1) becomes:(2)$$E\left(q,t\right)=\frac{\pi }{\lambda_{0}^{2}}E_{0}e^{-i\omega_{0}t}\frac{e^{ikR}}{R}\sin\left(\phi \right)\overset{N}{\underset{k}{\sum }}e^{iq\cdot R_{k}\left(t\right)}\underset{v_{k}}{\int }\Delta \varepsilon_{k}\left(r_{k},t\right)e^{iq\cdot r_{k}}dr_{k}\;$$
where now $\Delta \varepsilon_{k}\left(r_{k},t\right)=\varepsilon \left(r_{k},t\right)-\langle \varepsilon_{0}\rangle$ is the optical mismatch between the particles and the solvent, which is characterized by the average dielectric constant $\langle \varepsilon_{0}\rangle$. Note that the integral is not extended anymore to the entire scattering volume but to all the particle volumes $v_{k}$, with $e^{iq\cdot R_{k}\left(t\right)}$ being the time-dependent phase terms that depend on particle positions. By squaring Equation (2), we obtain the scattered intensity:(3)$$I\left(q,t\right)=\frac{\pi^{2}}{\lambda_{0}^{4}}\frac{I_{0}}{R^{2}}\sin^{2}\left(\phi \right)\overset{N}{\underset{k,j=1}{\sum }}a_{k}\left(q\right)a_{j}^{*}\left(q\right)e^{iq\cdot \left[R_{k}\left(t\right)-R_{j}\left(t\right)\right]}$$
where $a_{k}\left(q\right)=\int_{v_{k}}\Delta \varepsilon_{k}\left(r_{k},t\right){\;e}^{iq\cdot r_{k}}dr_{k}$ is the amplitude of the field scattered by the k-th particle.

A direct physical insight into Equation (3) can be gained if we make the simplifying assumption of homogeneous non-absorbing identical particles. Thus, if $\varepsilon_{1}$ indicates the (real) dielectric constant of the particles, $\Delta \varepsilon \left(r_{k},t\right)=\Delta \varepsilon =\varepsilon_{1}-\varepsilon_{0}$ and
(4)$$a\left(q\right)=\Delta \varepsilon \int_{v}e^{iq\cdot r}dr$$

By inserting Equation (4) into Equation (3), we get
(5)$$I\left(q,t\right)=\frac{A^{2}}{R^{2}}v^{2}I_{0\;}\sin^{2}\left(\phi \right)P\left(q\right)S\left(q,t\right)\;\left(\mathrm{identical}\;\mathrm{homogeneous}\;\mathrm{particles}\right)$$
where
(6a)$$A=\frac{\pi }{\lambda_{0}^{2}}\Delta \varepsilon$$
(6b)$$P\left(q\right)=\frac{1}{v^{2}}{\left|\int_{v}e^{iq\cdot r}dr\right|}^{2}$$
(6c)$$S\left(q,t\right)=\overset{N}{\underset{k,j=1}{\sum }}e^{iq\cdot \left[R_{k}\left(t\right)-R_{j}\left(t\right)\right]}$$
$P\left(q\right)$ is called *form factor* and is normalized to unity ($P\left(0\right)=1$), whereas $S\left(q,t\right)$ is called *structure factor* and is normalized so that $S\left(0,t\right)=N^{2}$. Equation (5) is valid under the so-called Rayleigh-Debye-Gans (RDG) approximation [1], which is the equivalent of the Born approximation for discrete scatterers. RDG approximation requires that (*i*): the optical mismatch Δε between particles and medium is small enough so that $\left(n_{1}/n_{0}-1\right)\ll 1$ ($n_{0}=\sqrt{\varepsilon_{0}}$, $n_{1}=\sqrt{\varepsilon_{1}}$) and (*ii*): the particle is “optically thin”, implying that phase difference between the light travelling in the medium and inside the particle is negligible. Quantitatively, the latter condition corresponds to $2\pi a\left(n_{1}-n_{0}\right)/\lambda_{0}\ll 1$, where a is the particle radius. Note that, when the particle is physically very small ($a\ll \lambda_{0}$), $P\left(q\right)\to 1$ for any q. This is so called Rayleigh (or dipole) scattering when all the sub-volumes dr oscillate in phase, and the particle behaves as a single dipole. Under these conditions, the scattered intensity distribution is isotropic and proportional to $\lambda_{0}^{-4}$.

In conclusion, Equation (5) describes the essence of the LS technique: the scattered intensity is proportional to the product $P\left(q\right)S\left(q,t\right)$ in which $P\left(q\right)$ provides information on particle size, structure, and morphology, whereas $S\left(q,t\right)$ depends on the particle’s motion and provides information on particle dynamics.

### 2.1. Static Light Scattering (SLS)

Static Light Scattering (SLS) is based on the measure of the time-averaged angular distribution of the light intensity scattered by a sample. Thus, Equation (5) must be averaged over a measuring time T much larger than the typical fluctuation time due to particle motions so that we can define $I\left(q\right)\equiv \langle I{\left(q,t\right)}_{T}\rangle$. Thus, under the assumption that the suspension is so dilute that particles do not interact, the averaging of Equation (5) reads:(7)$$\begin{array}{l} I\left(q\right)=\frac{A^{2}}{R^{2}}v^{2}I_{0}\sin^{2}\left(\phi \right)P\left(q\right)\left[N+\overset{N}{\underset{k\neq j}{\sum }}\langle e^{iq\cdot \left[R_{k}\left(t\right)-R_{j}\left(t\right)\right]}\rangle {}_{T}\right]\\ =\frac{A^{2}}{R^{2}}v^{2}I_{0}\sin^{2}\left(\phi \right)P\left(q\right)N\;\left(\mathrm{identical}\;\mathrm{noninteracting}\;\mathrm{homogeneous}\;\mathrm{particles}\right) \end{array}$$
where the sum term inside the square parenthesis vanishes because of statistical independence between positions of particles k and j. Thus, Equation (7) tells us that the time-average scattered intensity of a dilute suspension is simply given by the sum of the intensities scattered by all the particles inside the scattering volume.

Orientational averaging

It is worth pointing out that when the sample is made of a collection of randomly oriented anisotropic particles (such as ellipsoids, cylinders, platelets, etc.) Equation (5) must be averaged not only over time but also over all the possible orientations. For this purpose, we rewrite Equation (6b) as:(8)$$P\left(q\right)=\frac{1}{v^{2}}{\left|\int \phi \left(r\right)e^{iq\cdot r}dr\right|}^{2}$$
where the integral has been extended to the entire space, and we have introduced the local volume fraction $\phi \left(r\right)$, which is equal to 1 inside the particle and 0 outside, so that $\int \phi \left(r\right)dr=v$. The integral appearing in Equation (8) represents the Fourier transform of $\phi \left(r\right)$. Thus, by using a well-known property of the Fourier transform, we can equivalently rewrite Equation (8) as:(9)$$P\left(q\right)=\frac{1}{v^{2}}\int G_{\phi }\left(r\right)e^{iq\cdot r}dr$$
where
(10)$$G_{\phi }\left(r\right)=\int \phi \left(x\right)\phi \left(x+r\right)dx$$
is the spatial correlation integral of $\phi \left(x\right).$ $G_{\phi }$ is normalized so that $G_{\phi }\left(0\right)=v$ and $\int G_{\phi }\left(r\right)dr=v^{2}$. Equation (9) states that the form factor $P\left(q\right)$ is the Fourier transform of the (volume fraction) density-density correlation function. We can now average Equation (9) over orientations. By indicating with $G_{\phi }\left(r\right)\equiv \langle G_{\phi }{\left(r\right)\rangle }_{or}$, we obtain
(11)$$P\left(q\right)\equiv \langle P{\left(q\right)\rangle }_{or}=\frac{1}{v^{2}}\int G_{\phi }\left(r\right)e^{iq\cdot r}dr$$
and consequently
(12)$$P\left(q\right)=\frac{1}{v^{2}}\int 4\pi r^{2}G_{\phi }\left(r\right)\frac{\sin\left(qr\right)}{qr}dr$$
showing that the form factor is a function only of the modulus of the wavevector $q=\left|q\right|$. The function $p\left(r\right)=4\pi r^{2}G_{\phi }\left(r\right)$ is called the “pair distribution function” and is proportional to the probability density of finding two infinitesimal sub-volumes inside the particle at a distance r.

A clear physical insight into the meaning of $G_{\phi }\left(r\right)$ and $p\left(r\right)$ can be gained if we make the assumption that the particle is a rigid assembly of $N_{a}$ identical subunits or monomers of size a and volume $v_{a}$, so that $N_{a}v_{a}=v$. Thus the local volume fraction reads
(13)$$\phi \left(r\right)=v_{a}\overset{N_{a}}{\underset{i}{\sum }}\phi_{a}\left(r-r_{i}\right)$$
where $\phi_{a}\left(r\right)$ is the monomer volume fraction. If the monomer is so small that it can be considered a point-like particle (a≪λ), we have $\phi_{a}\left(r\right)=v_{a}\delta_{3}\left(r\right)$ and the overall volume fraction reads
(14)$$\phi \left(r\right)=\overset{N_{a}}{\underset{i}{\sum }}\delta_{3}\left(r-r_{i}\right)$$
where $\delta_{3}$ indicates the three-dimensional Dirac’s delta and $r_{i}$ are the atoms’ coordinates. By inserting Equation (14) into Equation (10), we get:(15)$$G_{\phi }\left(r\right)=v_{a}^{2}\overset{N_{a}}{\underset{i,j}{\sum }}\delta_{3}\left[r-(r_{i}-r_{j})\right]$$
and its orientational average reads:(16)$$G_{\phi }\left(r\right)=\frac{v_{a}^{2}}{4\pi r^{2}}\overset{N_{a}}{\underset{i,j}{\sum }}\delta_{1}\left(r-d_{ij}\right)$$
where $\delta_{1}$ indicates the one-dimensional Dirac’s delta and $d_{ij}=\left|r_{i}-r_{j}\right|$ is the distance between the *i*-th and *j*-th atoms. In Equation (16), we have used the identity $\delta_{3}{\left(r-R\right)}_{or}=\delta_{1}\left(r-R\right)/4\pi r^{2}$. Finally, by inserting Equation (16) into Equation (12), we obtain:(17)$$P\left(q\right)=\frac{1}{N_{a}^{2}}\overset{N_{a}}{\underset{i,j}{\sum }}\frac{\sin\left(qd_{ij}\right)}{qd_{ij}}$$
which is the famous Debye equation, worked out in 1915 [7] for the interpretation of X-ray powder diffraction data.

Low-angle regime

Whereas the form factor $P\left(q\right)$ depends on the overall structure and shape of the scattering particle, its profile at low q’s depends only on its average size. Indeed, by recalling that for x→0, $\sin\left(x\right)/x\sim 1-x^{2}/6$, we can expand Equation (17) in powers of q obtaining:(18)$$P\left(q\right)=1-\frac{q^{2}R_{G}^{2}}{3}+O\left(q^{4}\right)$$
where
(19)$$R_{G}^{2}=\frac{1}{N_{a}^{2}}\overset{N_{a}}{\underset{i\neq j}{\sum }}{\left(d_{ij}\right)}^{2}$$
is called the gyration radius of the particle. Equation (19) can be rewritten [8] as:(20)$$R_{G}^{2}=\frac{1}{N_{a}}\overset{N_{a}}{\underset{i}{\sum }}{\left|r_{i}-r_{c.m.}\right|}^{2}$$
where $r_{c.m.}$ is the center of mass of the particle. Thus, $R_{G}^{2}$ is a measure of the quadratic size of the particle, independent of its shape and structural morphology. Equation (18) is the basis of the so-called Guinier analysis [9], which provides a quantitative estimate of the particle size from the low-q behavior of the scattered intensity, without necessity of modelling its overall structure.

Absolute units

Equation (9) or its orientation averaged version, Equation (12), can be rewritten in absolute units, i.e., in terms of the scattered power per unit solid angle $\left(dP/d\Omega \right)=I\;R^{2}$. Thus, by indicating with $P_{0}$ the incident power and with L the length of the scattering volume, we obtain (see Appendix B):(21)$$\frac{dP\left(q\right)}{d\Omega }=R\left(q\right)\sin^{2}\left(\phi \right)P_{0}L$$
where
(22)$$R\left(q\right)=K_{opt}cMP\left(q\right)$$
is called the Rayleigh Ratio [${\mathrm{cm}}^{-1}$] and represents the time-averaged scattered power per unit solid angle, per unit incident power, and per unit length of the scattering volume. In Equation (22) c [${g\;\mathrm{cm}}^{-3}$] is the sample concentration, M [g] the particle’s molecular weight, and $K_{opt}$ [${\mathrm{cm}}^{2}\;g^{-2}$] an optical constant given by
(23)$$K_{opt}=\frac{1}{N_{A}}\frac{4\pi^{2}}{\lambda_{0}^{4}}n_{0}^{2}{\left(\frac{dn}{dc}\right)}^{2}$$
where $N_{A}$ is the Avogadro number, $n_{0}$ the refraction index of the solvent and dn/dc its increment with respect to the sample concentration. Since the latter two parameters are usually known (or easily measurable), $K_{opt}$ is easily determined. Typical values for $K_{opt}$ in the visible range are $\sim 5\times 10^{-7}\;{\mathrm{cm}}^{2}\;g^{-2}$ (polystyrene in organic solvents) or $\sim 4\times 10^{-7}{\mathrm{cm}}^{2}\;g^{-2}$ (proteins in aqueous solvents).

Equation (22) is the basis for the determination of the particle’s molecular weight M, which can be recovered from the zero-q extrapolation of $R\left(q\right)$.

Polydispersity

In the presence of sample polydispersity characterized by a number distribution $P_{N}\left(R\right)$ it is easy to show [3] that Equation (22) becomes
(24)$$R\left(q\right)=K_{opt}c{\langle M\rangle }_{wt}\langle P{\left(q\right)\rangle }_{z}$$
where
(25)$$\langle M_{w}\rangle =\frac{\int_{0}^{\infty }\left[P_{N}\left(R\right)M\left(R\right)\right]M\left(R\right)dR}{\int_{0}^{\infty }P_{N}\left(R\right)M\left(R\right)dR}$$
(26)$$\langle P{\left(q\right)\rangle }_{z}=\frac{\int_{0}^{\infty }\left[P_{N}\left(R\right)M^{2}\left(R\right)\right]P\left(q\right)dR}{\int_{0}^{\infty }P_{N}\left(R\right)M^{2}\left(R\right)dR}$$$\langle M_{w}\rangle$ is the weight-average molecular weight of the particles, averaged by using the weight distribution $P_{w}\left(R\right)=P_{N}\left(R\right)M\left(R\right)$. $\langle P{\left(q\right)\rangle }_{z}$ is the z-average form factor, averaged by using the z-distribution $P_{z}\left(R\right)=P_{N}\left(R\right)M^{2}\left(R\right)$. Note that the z-distribution is often called Intensity distribution because for very small particles ($a\ll \lambda_{0}$), the intensity is proportional to $M^{2}\left(R\right)$.

### 2.2. Dynamic Light Scattering (DLS)

Dynamic Light Scattering (DLS) is based on the determination of the translational diffusion coefficient D of particles freely moving in a fluid. This task can be tackled by measuring the normalized Intensity-Intensity auto-correlation function $g_{2}\left(q,\tau \right)=\langle I\left(q,t\right)I\left(q,t+\tau \right)\rangle /\langle I{\left(q\right)\rangle }^{2}$ of the light scattered by the sample at a given q. If the sample is made of a dispersion of a large number N of non-interacting particles undergoing a Brownian motion, the scattered electric field is described by a complex Gaussian stochastic process and $g_{2}\left(q,\tau \right)$ is related to the normalized field-field correlation function $g_{1}\left(q,\tau \right)=\langle E\left(q,t\right)E^{*}\left(q,t+\tau \right)\rangle /\langle I\left(q\right)\rangle$ by the Siegert relation
(27)$$g_{2}\left(q,\tau \right)=1+\beta {\left|g_{1}\left(q,\tau \right)\right|}^{2}$$
where β is known as the spatial coherence factor that depends on the number $N_{ca}$ of coherence areas detected by the collection optics ($\beta \sim 1/N_{ca}$ for $N_{ca}\gg 1$). Under the further ideal condition of a sample made of monodisperse particles, we have:(28)$$g_{1}\left(q,\tau \right)=\exp\left(-\tau /\tau_{c}\right)$$
(29)$$g_{2}\left(q,\tau \right)=1+\beta \exp\left(-2\tau /\tau_{c}\right)$$
where $\tau_{c}={\left(Dq^{2}\right)}^{-1}$ is the field-field correlation time. Thus $g_{2}\left(q,\tau \right)$ decays at a rate which is double with respect to $g_{1}\left(q,\tau \right)$ with a decay time equal to $\tau_{c}/2$. By fitting the intensity correlation data to Equation (29), one can recover D, and in turn, by using the Stokes-Einstein relation, the hydrodynamic diameter
(30)$$d_{h}=\frac{k_{B}T}{3\pi \eta D}$$
where $k_{B}$ is the Boltzmann constant, T the absolute temperature, and η the viscosity. Under typical working conditions in aqueous solvents (T=25C, η=0.01g/s cm), λ=532 nm, n=1.33, θ=90°, we have $q=22{\;\mu m}^{-1}$ and the relation between $\tau_{c}$ and $d_{h}$ is $\tau_{c}\left[s\right]=4.6394\times 10^{-6}d_{h}\left[\mathrm{nm}\right]$.

Polydispersity

In the presence of polydispersity, it is easy to show that Equation (28) becomes:(31)$$g_{1}\left(q,\tau \right)=\int_{0}^{\infty }P_{z}\left(\tau_{c}\right)\exp\left(-\tau /\tau_{c}\right)d\tau_{c}$$
where $P_{z}\left(\tau_{c}\right)$ is the Intensity- or z-weighted distribution of the correlation times $\tau_{c}$. Equation (31) is a linear integral equation, which is a classic example of an ill-posed problem. Thus, its solution (i.e., the recovery of $P_{z}\left(\tau_{c}\right)$) is not a trivial task and is commonly worked out by using iterative regularized inversion methods, such as the one based on the CONTIN algorithm is adopted in many commercial DLS instruments. However, due to the heavy ill-posedness of the problem together with the fact that $g_{1}\left(q,\tau \right)$ is not directly measured but recovered by inverting Equation (27), the solution of Equation (31) might be unreliable, as it may happen in the case of noisy $g_{2}$ data, or when the sample contains particles of very different sizes.

A partial alternative approach is to use the classical cumulants analysis [10], which, however, works only for narrow distributions ($\sigma_{\tau_{c}}/\langle \tau_{c}\rangle \ll 1$). By writing $\tau =\langle \tau_{c}\rangle +\delta \tau_{c}$, where $\langle \delta \tau_{c}\rangle =\int_{0}^{\infty }\delta \tau_{c}P_{z}\left(\tau_{c}\right)d\tau_{c}=0$, we can expand up to the second order the exponential term appearing in Equation (31) obtaining: (32)$$\exp\left(-\tau /\tau_{c}\right)=\exp\left(-\tau /\langle \tau_{c}\rangle \right)\left[1+\frac{\delta \tau_{c}}{\langle \tau_{c}\rangle {}^{2}}\tau +\frac{1}{2}\frac{\delta \tau_{c}{}^{2}}{\langle \tau_{c}\rangle {}^{4}}\tau^{2}\right]$$
by inserting Equation (32) into Equation (31) and in turn into Equation (27), by recalling that $\langle \delta \tau_{c}\rangle =0$ and maintaining only terms up to the second order, we get: (33)$$g_{2}\left(q,\tau \right)=1+\beta \exp\left(-2\tau /\langle \tau_{c}\rangle \right)\left[1+\frac{\delta \tau_{c}{}^{2}}{\langle \tau_{c}\rangle {}^{4}}\tau^{2}\right]$$
which depends on $\langle \tau_{c}\rangle$ and $\sigma_{\tau_{c}}$. Thus Equation (33) can be used as a fitting function for recovering the z-average decay time and standard deviation of the distribution that characterizes the (narrow) sample polydispersity.

### 2.3. Multi Wavelength Turbidimetry (MWT)

The technique MWT is based on the measurement of the power $P_{T}\left(\lambda_{0}\right)$ transmitted by a sample that is illuminated with a white source of incident power $P_{0}\left(\lambda_{0}\right)$. For an absorbing and scattering sample, under the assumption that there is no multiple scattering, $P_{T}$ and $P_{0}$ are related by the Lambert-Beer law [11]
(34)$$P_{T}\left(\lambda_{0}\right)=P_{0}\left(\lambda_{0}\right)e^{-\tau \left(\lambda_{0}\right)L}$$
where L is the cell optical path and $\tau \left(\lambda_{0}\right)$ is the extinction coefficient, customarily measured in ${\mathrm{cm}}^{-1}$. In the case of non-absorbing samples or in the spectral regions where there are no absorption bands, the light extinction is only due to scattering. In this case, $\tau \left(\lambda_{0}\right)$ takes the name of turbidity coefficient that can be computed by integrating Equation (22) over the entire solid angle dΩ: (35)$$\tau \left(\lambda_{0}\right)=\int_{4\pi }R\left(q\right)\sin^{2}\left(\phi \right)d\Omega$$
where $\sin^{2}\left(\phi \right)=1-\sin^{2}\left(\theta \right)\cos^{2}\left(\varphi \right)$ and $d\Omega =\sin\left(\theta \right)d\theta d\varphi$, being θ and φ the common polar and azimuthal angles used in spherical coordinates. Note that the polar angle θ coincides with the scattering angle, being the z axis defined as the direction of the incident beam. Equation (35) can be conveniently rewritten (after some calculation) as: (36)$$\tau \left(\lambda_{0}\right)=8\pi \int_{0}^{1}R\prime \left(x\right)x\left[1-2x^{2}+2x^{4}\right]dx$$
where $x=q/q_{max}=\sin(\theta /2)$ and $R\prime \left(x\right)=R\left[x\left(q\right)\right]$. As an example, we can compute the turbidity of Rayleigh scatterers ($a\ll \lambda_{0}$) for which $P\left(q\right)=1$ and consequently $R\left(x\right)=K_{opt}cM.$ The integration of Equation (36) gives $\tau \left(\lambda_{0}\right)=\left(8\pi /3\right)K_{opt}cM$, implying that if one neglects the (typically known) lambda dependence of $n_{0}$ and dn/dc, the turbidity scales as $\tau \left(\lambda_{0}\right)\sim \lambda_{0}^{-4}$. This power-law behavior with an exponent of −4 is specific to Rayleigh (or point-like) scatterers and provides no information about the size and morphology of the scatterers.

Conversely, when the size of the scatterers becomes comparable to or larger than $\lambda_{0}$ quite useful information about the particle structure and morphology can be extracted from the MWT technique. For example, for very long and thin rigid, straight cylinders of length L and diameter d ($d\ll \lambda_{0}\ll L$), the form factor is $P\left(q\right)=\pi /qL$ leading to
(37)$$\tau \left(\lambda_{0}\right)\;\sim \;n_{0}{\left(\frac{dn}{dc}\right)}^{2}c\mu \lambda_{0}^{-3}$$
where μ is the mass/length ratio of the cylinder, which provides information on the cylinder cross-section ($\mu =\left(\pi /4\right)\rho d^{2}$, with ρ being the cylinder density). Equation (37) is the basis of the so-called Carr-Herman method [12], commonly used for the characterization of macromolecular solutions of linear polymers.

Equation (37) can be generalized to the case of large fractal aggregates (or clusters) made of an assembly of many small monomers [13,14]. Under the assumption that the cluster gyration radius $R_{G}$, the monomer diameter d and the wavelength fulfill the condition $d\ll \lambda_{0}\ll R_{G}$, the form factor scales as $P\left(q\right)\sim {\left(qR_{G}\right)}^{-D_{m}}$ leading to
(38)$$\tau \left(\lambda_{0}\right)\;\sim \;n_{0}{\left(\frac{dn}{dc}\right)}^{2}c\rho d^{1-D_{m\;}}\lambda_{0}^{-4+D_{m}}$$
where $D_{m}$ is the mass fractal dimension that characterizes the structural morphology of the aggregate. Therefore, Equation (38) shows that the MWT technique can be quite useful for investigating the structural properties of fractal systems.

## 3. Materials and Methods

The two instruments described in this section and used for the measurements reported below have been realized by combining homemade and commercial mechanical, optical and electronic components.

### 3.1. LA-SLS + MWT Instrument

The apparatus for the coupled Low Angle Static Light Scattering (LA-SLS), and Multi Wavelength turbidimetry (MWT) consists of a homemade LA-SLS setup, which has been implemented with a commercial fiber optic spectrophotometer for MWT measurements in the IR-VIS-UV range. As shown in Figure 1a, by using two PC-controlled shutters, the sample cell can alternatively be illuminated with a focused laser beam (10 mW, λ=532 nm) or with a collimated white source beam. The LA-SLS sensor (Figure 1b) is made of 31 annular quarters of photodiode rings centered around the optical axis, where a 200 μm pinhole allows the beam to pass clear. The detector is placed on the Fourier plane, i.e., the plane where the beam is focused. On this plane, each radial position r corresponds to the same scattering angle via the relation $\theta =tg^{-1}\left(r/z\right)$, where z is the distance between the cell and the sensor. Thus, since the minimum and maximum radii of the rings are $r_{min}=180\;\mu m$ and $r_{max}=17\;\mathrm{mm}$, the range of the detectable scattering angles is ~0.1–10° (z=100 mm).

The MWT setup is based on the use of two identical commercial fiber optics spectrophotometers (Ocean Optics, mod. HR2000+). The sample cell is illuminated by a white light source (Ocean Optics, mod. DH2000) made of a deuterium and a halogen lamp coupled together onto a 600 μm core optical fiber, which is then split into two 200 μm core optical fibers by a 50–50 fiber-optics beam splitter. The light from one fiber is sent directly to the reference spectrophotometer, which is used for normalizing the turbidimetry data as usually done in a standard double-beam spectrophotometer. The light from the other fiber is collimated and shined onto the sample, approximately in the same spot where the laser beam hits the sample. The light beam, after having passed through the sample, is reflected by a mirror placed in the blind zone of the Fourier plane (the one not occupied by the LA-SLS sensor) onto a collecting lens (identical to the collimating one), that is coupled to a 600 μm core optical fiber, whose end face is placed in the focal plane of the lens. In this way, the fiber core acts as a spatial filter collecting only the transmitted light and rejecting most of the light scattered at low angles [15]. A detailed description of the LA-SLS+MWT setup can be found in Ref. [14].

### 3.2. WA-SLS and DLS Instrument

The Wide Angle (WA-SLS) and DLS instrument is a non-conventional LS photometer capable of performing both SLS and DLS measurements. This instrument was developed a long time ago at the University of California, Santa Barbara, by the group of D.S. Cannell [16] and recently donated to our laboratory. Differently from most commercial LS instruments that work by using a mechanical goniometer for changing the angles at which the scattered light is collected, this photometer employs 18 fixed angles, a feature that guarantees high mechanical and optical stability. As sketched in Figure 2, the instrument works by shining a mildly focused 25 mW CW He-Ne Laser operating at λ=632.8 nm onto a cylindrical cell placed at the center of a round tank filled with dust-free water. The scattered light coming from the central portion of the beam is collected by 18 lenses (placed on the side of the tank) and brought to the entrance face of 18 optical fibers. All fibers are then bundled together onto the photocathode of a photomultiplier. A set of 18 PC-controlled shutters placed in front of each fiber allows the opening of one channel at a time, ensuring that intensities scattered at the various angles do not mix up onto the detector. The angular range covered by this instrument is ~3−160°. Since the optical fibers used in this instrument collect a few coherence areas, each channel can operate both in the static and dynamic mode. A detailed description of this instrument can be found in Ref. [16].

## 4. Results

In this section, we report three examples of applications of the LS techniques for characterizing non-stationary systems, such as aggregation of colloidal suspensions, aggregation of hydroxyapatite nanoparticles, and formation of fibrin gels.

### 4.1. Colloidal Aggregation (LA-SLS and MWT)

The colloidal aggregation of small monomers (such as gold, silica, or polystyrene latex nanospheres) represents a paradigmatic example of how the LS techniques can be fruitfully exploited for characterizing the kinetics of non-stationary systems undergoing an irreversible growth process.

Water-based colloidal suspensions are usually stabilized against aggregation by the presence of charges on the surface of the colloidal particles. Indeed, the interaction potential between colloidal particles can be schematized as the sum of two opposing terms: short-range Van der Waals attraction and long-range screened Coulomb repulsion. In normal conditions, repulsion is stronger than attraction, but upon the addition of salt to the suspension, the surface charges are screened out, and particles brought to contact by Brownian motion can irreversibly stick together and start forming aggregates (or clusters).

When the activation barrier is so low that every particle’s (or clusters’) encounter leads to an irreversible sticking, the aggregation process is termed Diffusion Limited Cluster Aggregation (DLCA), indicating that the aggregation process is limited by the clusters’ diffusion. Under DLCA conditions, the cluster size distribution is expected to be rather monodisperse, the cluster mass M grows linearly with time (M~t), whereas the cluster gyration radius $R_{G}$ grows as a power law ($R_{G}\sim t^{1/D_{m}}$), where $D_{m}$ is the cluster mass fractal dimension. On the contrary, in the presence of a small but non-negligible activation barrier, two particles or clusters may need to meet many times before sticking to each other, and the process is said to be a Reaction Limited Cluster Aggregation (RLCA). Clusters grown under RLCA conditions are characterized by broad distributions, with both $M\;\mathrm{and}\;R_{G}$ growing exponentially in time. In both cases (DLCA and RLCA), the structure of the aggregates exhibit a fractal morphology implying that M scales with $R_{G}$ according to
(39)$$M\;\sim \;m_{a}{\left(\frac{R_{G}}{a}\right)}^{D_{m}}$$
where a is the monomer radius and $m_{a}$ its mass. Under the conditions of DLCA or RLCA, most of the structural features of the fractal aggregates do not depend on the microscopic and/or chemical details of the colloid system, and, therefore, fractal aggregation behaves like a universal process [17]. DLCA aggregates have a fractal dimension $D_{m}\sim 1.7$, independent on the fact that the colloidal particles are made of gold, polystyrene or silica. In a similar way, RLCA leads to more compact objects having dimension $D_{m}\sim 2.1$. Figure 3 shows a pictorial description of the growth of a fractal colloidal aggregate.

SLS is a perfect tool for studying the fractal morphology of a fractal aggregate grown from polystyrene latex nanospheres dispersed in water. Indeed, their density ($1.05\;g/{\mathrm{cm}}^{3}$) matches quite closely the density of water and, therefore, fairly large monomers (~100 nm), providing a high scattering signal, and can be used as building blocks of the aggregating cluster without sedimentation effects altering their aggregation kinetics.

In our experiment, we used latex spheres 70 nm in diameter (Thermo-Fisher Scientific Co.) at a concentration number $c_{0}=5.6\times 10^{10}{\;\mathrm{cm}}^{-3}$ (volume fraction concentration $=10^{-5}$), and the aggregation was induced by adding the divalent salt $\left[{\mathrm{MgCl}}_{2}\right]=15\;\mathrm{mM}$. The scattered intensities taken at different times after the addition of salt are shown in Figure 4 (open symbols) as a log-log plot. The figure shows the typical behavior expected for the evolution of the scattered intensities in a colloidal aggregation experiment: a strong (~2.5 decades) increase in the zero-q scattered intensity, accompanied by a remarkable change in the shape of $I\left(q\right)$, with the curve roll-off moving towards small q and the large-q data laying on the same asymptote. The latter one is the signature of the aggregate’s fractal morphology and represents a measure of their mass fractal dimension $D_{m}$ because asymptotically (q→∞) → $I\left(q\right)\sim q^{-D_{m}}$. The data in Figure 4 were fitted to the so-called Fisher-Burford function [19].
(40)$$I\left(q\right)=\frac{I\left(q=0\right)}{{\left[1+\frac{2}{3D_{m}}q^{2}R_{G}^{2}\right]}^{\frac{D_{m}}{2}}}$$
in which the fitting parameters were the zero-q intensity $I\left(q=0\right)$, the fractal dimension $D_{m}$ and the cluster gyration radius $R_{G}$. The fittings, reported in Figure 4a as solid curves, are quite satisfactory and allow to estimate $D_{m}=1.62\pm 0.02$, a figure that, although somewhat smaller than expected, is consistent with a DLCA growth modality.

A further check that the aggregation followed a DLCA growth modality was found by studying the time behavior of M and $R_{G}$, which were recovered from data fitting (note that $I\left(q=0\right)$ is proportional to M, see Equation (22)). Indeed, Figure 5a shows that $R_{G}$ grows as a power-law characterized by an exponent of 0.62, which is consistent with $D_{m}=1/0.62=1.62$. Similarly, Figure 5b shows that M grows linearly with time, as expected for DLCA. Figure 5c shows that the fractal dimension of the clusters is consistent with $D_{m}\sim 1.62$ during the entire aggregation process, but its recovery becomes rather accurate only when the clusters grow to sizes $R_{G}≳5\;\mu m$. Finally, Figure 5d shows that the scaling between $M\;\mathrm{and}\;R_{G}$ follows Equation (39), which is a key signature of the clusters’ fractal morphology.

The growth kinetics of a colloidal aggregation process can also be fruitfully investigated by using the MWT technique implemented via commercial fiber optics spectrophotometers, as described in Section 3.1. Figure 4b reports the behavior of the sample turbidity as a function of the wavelength $\lambda_{0}$ for different times after addition of salt. The sample was the same as Figure 4a. The figure shows that at the beginning when the suspension is made of single monomers, the turbidity is relatively low and scales as $\tau \left(\lambda_{0}\right)\sim \lambda_{0}^{-4.4}$. Afterward, its amplitude constantly increases, and the decay exponent grows up to the final behavior $\tau \left(\lambda_{0}\right)\sim \lambda_{0}^{-2.8}$.

As described in Section 2.3, we can use the behavior of $\tau \left(\lambda_{0}\right)$ for estimating the mass fractal dimension of the aggregates via Equation (38), but this method apparently leads to a wrong value of $D_{m}=4-2.8=1.2$. However, due to the wavelength dispersion of the refraction indexes of both polystyrene and water [14,15], the exponent −4 expected from Rayleigh scattering is modified into an effective exponent of −4.4. Thus, from the slope measured at the latest time, we recover a value of $D_{m}=4.4-2.8=1.6$, which is in excellent agreement with the fractal dimension recovered from the LA-SLS measurements.

### 4.2. Hydroxyapatite Polymerization (SLS at Wide Angle + DLS)

Hydroxyapatite $\left({\mathrm{Ca}}_{10}{\left({\mathrm{PO}}_{4}\right)}_{6}{\left(\mathrm{OH}\right)}_{2}\right)$ nanoparticles (nHAs) are among the main constituents of hard tissues in living organisms, such as shell, bone, and teeth [20]. The size, morphology, and crystalline structure of nHAs determine the mechanical and biofunctional properties of these tissues, opening the way to exploit synthetic biomimetic nHAs in many biomedical and biotechnological applications, from regenerative medicine for bone repair and growth to tissue engineering and drug delivery [21]. However, the spectrum of nHAs applications goes well beyond the field of biomedicine. For example, nHAs are used as coatings, catalysts, water purification, chromatography, and, in general, in the field of material science, for the design and fabrication of new composite nHA-based materials with high mechanical strength and elasticity, similar to those of bone [21]. Very recently, the use of (properly doped) nHAs and congeners have also been proposed in the field of green agriculture as non-toxic, low-cost, and efficient nanofertilizers [22,23].

One of the still highly debated issues about nHAs is related to their crystallization mechanism and, in particular, how the growth kinetics and the structural-morphological properties of the final nHA nanoparticles depend on the chemical-physical properties of their starting solutions and intermediate nanosized solid precursors. Indeed, precipitation of poorly soluble calcium orthophosphates is highly dependent on precursor concentrations, medium acidity, temperature, the presence of complexing agents, and ionic strength. Altogether, these conditions drive the formation of different amorphous calcium phosphate (ACP) precursors, later transforming into nHAs. While kinetics plays a fundamental role in controlling such transformation, the thermodynamic stability of (bulk) apatite makes the formation of nHAs highly probable if enough time is allotted for ion exchange and nanoparticle maturation. These aspects have been beautifully discussed in a number of review papers [24,25] and books [26,27].

One of the most established mechanisms leading to nHAs during precipitation from an aqueous solution is the pristine formation of tiny amorphous calcium phosphate (ACP) nanoparticles, which, upon maturation (within minutes or hours if the reaction is kept at a physiological temperature of 37 °C) slowly transform into nHAs. How fast ACP transforms into nHAs when the ion concentrations are kept very low and complexing agents (such as citrate) are added to follow the process en relenti, is still an underexplored field of study, which we then tackled by the abovementioned WA-SLS and DLS techniques.

Specifically, nAPs were synthetized by mixing three solutions: (a) calcium chloride $\left({\mathrm{CaCl}}_{2,}\;0.2\;M\right),$ (b) sodium carbonate + potassium phosphate $\left({\mathrm{Na}}_{2}{\mathrm{CO}}_{3},\;0.1\;M+{\mathrm{HK}}_{2}{\mathrm{PO}}_{4},\;0.12\;M\right)$ and (c) citrate (tribasic) $\left(C_{2}H_{5}{\mathrm{Na}}_{3}O_{7},\;0.2\;M\right)$. The three solutions were mixed in equal volumes (40 μL) and diluted in water up to a final volume of 12 mL, producing a final nHAs concentration of $6.7\times 10^{-5}\;M$.

Upon mixing, we observed a slow but constant increase in the scattering signal accompanied by the formation of growing fractal clusters, as monitored both by SLS and DLS. Figure 6 shows the time evolution of the scattered intensity distribution from the beginning of aggregation when $I\left(q\right)$ is almost flat with a slight curvature at high q, up to ~21 h, when it attains a full power-law decay behavior, $I\left(q\right)\sim q^{-D_{m}}$ over the entire q-range. Thus the data in Figure 6 resemble very closely the behavior reported in Figure 4a for the growth kinetics of fractal colloids. Similarly, the $I\left(q\right)$ data were nicely fitted with the same fitting function (Equation (40)) obtaining for the fractal dimension $D_{m}=1.88\pm 0.03$, which is a figure in between the DLCA and RLCA expected fractal dimensions.

As done for the colloidal aggregation experiment, we can analyze the time evolution of the various cluster parameters for distinguishing between the two growth modalities. Figure 7a,b (lin-log plots) show that both $M\;\mathrm{and}\;R_{G}$ exhibit an exponential growth typical of an RLCA aggregation process. Figure 7c shows that the determination of $D_{m}$ becomes accurate only when the clusters grow to sizes $R_{G}≳100\;\mathrm{nm},$ and finally, Figure 7d confirms that the entire data analysis is self-consistent because the power-law scaling (Equation (39)) between $M\;\mathrm{and}\;R_{G}$ is characterized (at later times) by the exponent 1.88, which matches quite accurately the value of $D_{m}$ found from the asymptotic decay of scattered intensity distribution, namely $I\left(q\right)\sim q^{-D_{m}}$. To summarize, Figure 7 suggests that the aggregation process is consistent with an RLCA growth modality, in spite of the fact that the value found for $D_{m}$ is intermediate between DLCA and RLCA. This is not so surprising because, although universal values for the fractal dimensions have been proposed (and experimentally confirmed for ideal systems such as small colloids undergoing irreversible aggregation), in many situations, the conditions for a pure DLCA or RLCA do not exist, or they do change during the course of time. Thus intermediate regimes in which a transition RCLA → DLCA may be observed [28] or a restructuring of the cluster morphology leading to a transformation DCLA → RLCA may take place [29,30]. In all these situations, $D_{m}$ can assume values between DLCA and RLCA.

The results of Figure 6 and Figure 7 obtained via the WA-SLS technique can be compared (and possibly) validated by means of the DLS measurements taken on the same nHAs sample. Figure 8 shows such a comparison, where the $R_{G}$ data taken with WA-SLS are reported as equivalent diameter, i.e., $d_{h}=2\sqrt{5/3}R_{G}$ (solid black circles), whereas the $d_{h}$ recovered with DLS at three different angles are reported as open symbols. As one can notice, the DLS data are all consistent with each other and consistent with the static data only at the beginning of the aggregation process, when $d_{h}\sim 100–200\;\mathrm{nm}$. Later on, the DLS sizes become progressively smaller than the WA-SLS sizes, suggesting that the solvent can drain away from the loose and flexible structure of the clusters, resulting in an effective hydrodynamic radius smaller than its equivalent gyration radius. As for kinetics, both data sets are compatible with exponential growth (although at different rates), suggesting once again that the aggregation process is consistent with an RLCA growth modality.

### 4.3. Fibrin Polymerization (SLS at Low Angle and Wide Angle + MWT)

Fibrin gels are biological networks of fundamental importance in the process of blood coagulation [31,32]. Indeed, following an injury, they start to grow around the damaged tissue forming a network that traps platelets and other blood components, eventually forming the blood clot that stops bleeding. Fibrin gels also play important roles in other pathological and physiological situations, such as thrombosis and cancer [33], but they are also used in many biotechnological applications, from surgery (adhesives and sealants called fibrin glues) to tissue engineering and drug delivery. A thorough review on fibrin gels and their biotechnological applications can be found in [34].

Fibrin gels are grown from the polymerization of the macromolecule fibrinogen (FG) after activation by the enzyme thrombin (THR). According to the classic theory of fibrin formation [34], the activated monomers aggregate, forming half-staggered, double-stranded fibrils that initially grow in length and, only when they are long enough to interact with each other, start to branch and aggregate laterally and eventually a 3D network is obtained. The polymerization kinetics and the final structure of the gel depend on the physical-chemical conditions of the solution in which the gel is grown, such as the FG and THR concentrations, pH, ionic strength, or presence of Ca^++^ ions [35,36,37,38]. As a consequence, kinetics and structure are intimately related to each other, implying that the growth modality determines the final aged gel structure. The latter one depends on physical properties (such as fiber diameter and length, fiber elasticity and pore size) that are directly linked to their mechanical and biological functions [39]. For example, the fibers of a fibrin gel are very soft (if compared with equal diameter fibers of other biopolymer networks such as F-actin or collagen) and can be deformed or stretched to a quite large extent [40], a feature which is essential for the gel functioning as an efficient hemostatic plug and a wound healing matrix.

Figure 9a shows a typical rendering obtained from a stack of confocal images taken on an aged fibrin gel: the network is made of a collection of randomly oriented straight fibers joined together at some nodal points and separated by an average distance that is much larger than their diameters [41,42]. The fibers are almost monodisperse in size, but their density is not uniform, giving rise to correlated spatial density fluctuations that are characterized by a long-range spatial order, whose length scale is comparable with the distance between fibers.

Early LA-SLS and WA-SLS studies [35,36,37,41] have shown that the structure of these gels can be modeled as an assembly of densely packed fractal “blobs” of mass fractal dimension $1<D_{m}<2,$ size ξ, placed at an average distance $\xi_{0}\sim \xi$. Each blob, which corresponds to the regions of higher density in Figure 9a (yellow ellipses), is made of different-length straight fibers of diameter d, density ρ, joined at randomly distributed nodal points. We were also able to demonstrate that the parameter $\xi_{0}$ gives a correct estimate of the gel mesh or pore size, which can be quantitatively defined as the average diameter of the largest spheres that can be accommodated in the pores of the gel and are tangent to the surrounding fibers (see Figure 9b).

Static light scattering techniques can provide reliable estimates of all the above parameters $\left(D_{m},\;\xi_{0},\;d,\varrho \right)$, but this is a rather difficult task because of the huge range of length scales to be probed, which goes from fiber diameters (d~50–100 nm) to gel pore sizes ($\xi_{0}\sim 10\;\mu m$ or larger). Thus, a very wide range of q-vectors must be accessed, a requirement that can be accomplished by coupling LA-SLS and WA-SLS techniques.

In the past, we have performed this task [35], but always independently, i.e., by using different instruments that operate on different specimens (although prepared under the same conditions), different scattering cells, different laser sources, etc. The consequences are a rather low reproducibility and reliability of the results, also deepened by the fact that biological samples are themselves prone to be somewhat irreproducible. A way out of this problem is to perform measurements with different techniques on the same specimen at the same time, as done with the LA-SLS+MWT setup in Figure 1 [14].

Figure 10 reports log-log plots of LA-SLS and MWT data (and corresponding fits) taken on a polymerizing fibrin solution (FG 0.45 mg/ml in TBE, THR:FG molar ratio 1÷100) at different, relatively long times after adding thrombin to the FG solution. As shown, both the LA-SLS and MTW signals increase with time up to t~1000 s, when the gel attains its steady-state structure.

The data were fitted with two functions (developed according to the blob model described above, see Ref. [14]). For $R\left(q\right)$ we used the fitting function given by Equation (C.1) reported in the Supporting Information of Ref. [41] (with errata), whereas for $\tau \left(\lambda_{0}\right)$ we used the numerical integration given by Equation (36). Both functions depend on the four parameters $D_{m}$, $\xi_{0}$, d and ϱ, some of which are highly correlated if recovered from individual fits. For example, from LA-SLS we can reliably recover $D_{m}$ (from the high-q region slope) and $\xi_{0}$ (from the peak position), but not d and ϱ because the scattering amplitude depends on both ($I\sim \varrho d^{2}$). At the same time, the slope of the power-law behavior of the MWT depends not only on $D_{m}$ but also on d and to a lesser extent on $\xi_{0}$ because the conditions of Equation (38) are only partially fulfilled. Similarly the amplitude of MWT depend on d and ϱ.

We solved the problem by adopting a global fitting strategy, in which LA-SLS and MWT data are fitted simultaneously; this means that the two data sets are fitted with the two fitting functions described above, both controlled by the same floating parameters, namely $D_{m}$, $\xi_{0}$, d and ϱ. In this case, the $\chi^{2}$ minimization is carried out by taking into account the (weighted squared) deviations of each data set from its corresponding fitting function. Equivalently, the combined fitting of the two data sets can be performed by following a two-step procedure in which we first fit LA-SLS data and recover $D_{m}$ and $\xi_{0}$; then, we fit the MWT data (by keeping fixed $D_{m}$ and $\xi_{0}$ to the values found above) and recover d and ϱ. We have shown [14] that the two methods (global and two-step procedure fits) are fairly equivalent, with the first one being more robust because not affected by systematic errors introduced by the fixed parameters used in the two-step procedure. On the other hand, the second method is somewhat more flexible because the fixed parameters $D_{m}$ and $\xi_{0}$ can be estimated by using different techniques (other than LA-SLS), such as confocal microscopy [42] and rheometry [43].

By applying the two-step procedure to the data in Figure 10 corresponding to the aged gel (t>1000 s), we obtained $D_{m}=1.37\pm 0.01$, $\xi_{0}=13.2\pm 0.1\;\mu m$, d=90±5 nm, and $\rho =0.44\pm 0.03\;g/{\mathrm{cm}}^{3}$.

As to the kinetic aspect of the data presented in Figure 10, one may notice that, in the time range of the figure, both LA-SLS and MWT data exhibit a remarkable growth of their amplitudes, with only slight changes in their shape distributions. This feature implies that the fibrin network has already formed and attained its final structure after ~230 s. Later on, up to the final aged gel formation, the growth kinetics consist mainly of an increase in the fibers’ mass/length ratio (which is responsible for the amplitude increase), with almost no change in the gel pore size $\xi_{0}$ (constant peak position, Figure 10a) and the fractal dimension $D_{m}$.

The data in Figure 10 are consistent with a very simple growth mechanism: initially, the fibrinogen monomers polymerize into double-stranded linear fibrils [35] until these are so long that they interact and start linking to each other. At this time, the onset of gelation takes place, and the scaffold for the building up of the final gel network is outlined. As shown in Ref. [35], the fibrinogen concentration necessary for building up the scaffold is only a small fraction of the overall fibrinogen concentration (~10–20%) so that all the remaining free monomers or fibrils can diffuse around and bind to the frozen fibers that grow thicker and thicker (or denser and denser, or both) until no monomers are around. Thus, during this *thickening phase*, the gel structure remains unchanged, and the scattering amplitude constantly increases without almost no change in shape.

Noticeably, this growth mechanism appears to be quite suitable for letting the fibrin gel accomplish its main physiological task very efficiently, i.e., the process of blood coagulation. Indeed, a network made of thin fibrils is formed rather quickly and starts to trap blood components, followed by a phase in which the network strengthens its fibers (making them thicker) so that it can withstand the blood pressure when the blood clot is fully formed, and there is no more bleeding. During the entire process, the fibers need to be rather soft and very elastic (so as to bear local stresses and avoid breakage), a feature that is guaranteed by their low density ($\rho \sim 0.4\;g/{\mathrm{cm}}^{3}$), implying that the fibrils inside the fibers are not densely packed but intertwined with solvent molecules.

The growth model described above was further refined by carrying out coupled measurements of Small Angle X-ray Scattering (SAXS) and WA-SLS data on the *early phases* of the polymerization process (data not reported, see [38]) where we found out that the branching also takes place at the level of single fibrils. Thus, the formation of the initial scaffold is even more rapid, and the thickening phase also includes the collapse of the branched fibril on the frozen fibers.

## 5. Conclusions

In this work, we have reviewed and discussed the main features of the classical light scattering techniques, such as Static Light Scattering (SLS), Dynamic Light Scattering (DLS), and Multi Wavelength Turbidimetry (MWT). We have shown how these techniques can be very useful for characterizing the structure and the dynamics of various complex systems, both in their stationary states at thermodynamic equilibrium and when they undergo irreversible aggregating, polymerizing, or self-assembling kinetics. Three examples were reported:

Aggregation of 70 nm latex colloids that, when the suspension is destabilized by the addition of salt, aggregate to form fractal clusters characterized by a Diffusion Limited Cluster Aggregation (DLCA) growth kinetics. The final cluster was $R_{G}\sim 20–30\;\mu m$ in size with a fractal dimension with $D_{m}\sim 1.62$. This study was performed by using the LA-SLS + MWT techniques.Polymerization of hydroxyapatite nanoparticles starting from solutions of calcium chloride, potassium phosphate, and citrate reagents. In this case, the kinetics follow a Reaction Limited Cluster Aggregation (RLCA) growth modality, with aggregates as large as $R_{G}\sim 1–2\;\mu m$, characterized by a final fractal dimension $D_{m}\sim 1.88$. This study was performed by using the WA-SLS + DLS techniques.Formation of fibrin gels starting from the polymerization of a fibrinogen solution destabilized by the addition of thrombin. The late stages of this kinetic process revealed that the growing mechanism consists of a quick formation of a network made of thin fibrils, followed by a thickening phase during which the fibers become thicker and thicker and make the network increasingly stronger to withstand the blood pressure. This study, which was performed by using the LA-SLS + MWT techniques, provided a clear example of the benefits of coupling together different techniques. Indeed, by combining LA-SLS + MWT, we were able to recover the four parameters characterizing the gel structure (pore size, fractal dimension, fiber size, and density), which could not be recovered from single-methods data analysis.

Finally, we would like to point out that in this mini-review, we (deliberately) reported only some examples taken from our research group activity. We are perfectly aware that these examples are quite limited, and they are not at all fully representative of the wide spectrum of experimental implementations and applications of the LS techniques. However, thanks to them, we had the possibility of entering into the details of the various methods and discussing both their potentialities and limitations. Similarly, when presenting the advantages of coupling different methods, we limited ourselves to only LS techniques, without discussing the possibility of combining LS with SAXS and SANS techniques, but only mentioning an application [38]. This example already highlights that the combined use of LS and SAXS/SANS techniques would clearly enlarge the types of samples (and the q-range) investigable with scattering techniques, but this was beyond the purpose of this article.

## Figures and Tables

**Figure 1 nanomaterials-12-02214-f001:**
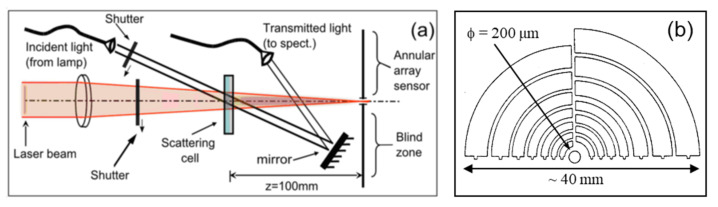
(**a**) Schematic diagram of the LALS + MWT setup. (**b**) Schematic diagram of the annular array sensor used for the LA-SLS measurements. Figure adapted from Ref. [14].

**Figure 2 nanomaterials-12-02214-f002:**
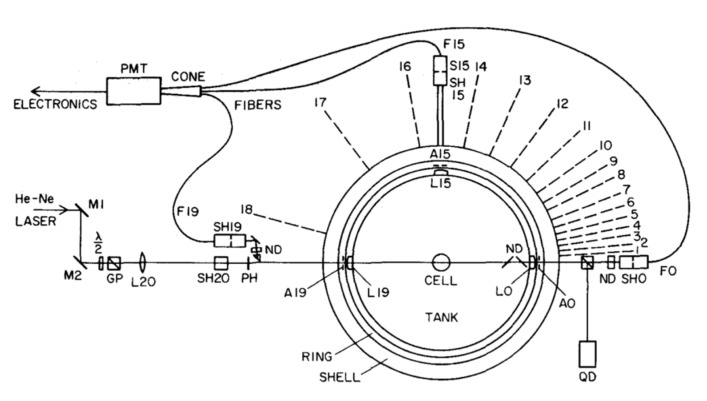
Sketch of the optical setup used in the WA-SLS instrument: the scattered light is collected at 18 fixed angles and measured, one at a time, by a single PMT. Figure reproduced with permission from Ref. [16].

**Figure 3 nanomaterials-12-02214-f003:**
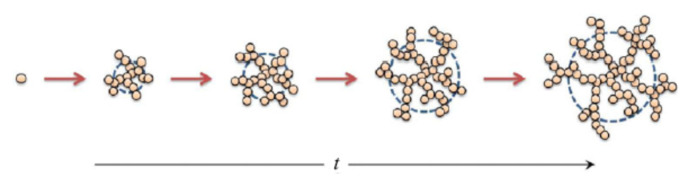
Pictorial description of the growth of a fractal colloidal aggregate. The mass M and the gyration radius $R_{G}$ of the aggregate both increase with time, and they scale as $M\sim {\left(R_{G}\right)}^{D_{m}}$, where $D_{m}$ is the mass fractal dimension of the aggregate. Picture reproduced with permission from Ref. [18].

**Figure 4 nanomaterials-12-02214-f004:**
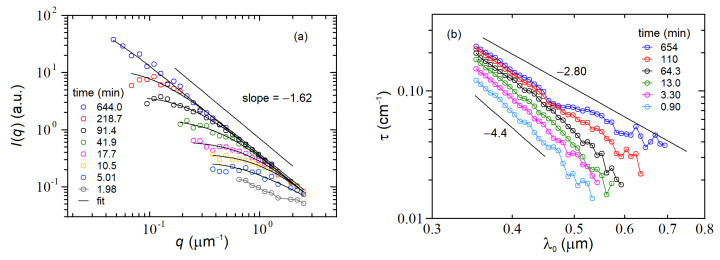
Colloidal aggregation of latex spheres characterized by the LA-SLS+MWT instrument. (**a**) Scattered intensity $I\left(q\right)$ data (symbols) and fits (continuous lines) as a function of q for different times after addition of salt to the colloidal suspension; (**b**) behavior of the turbidity as a function of wavelength at different times for the same sample of panel (**a**).

**Figure 5 nanomaterials-12-02214-f005:**
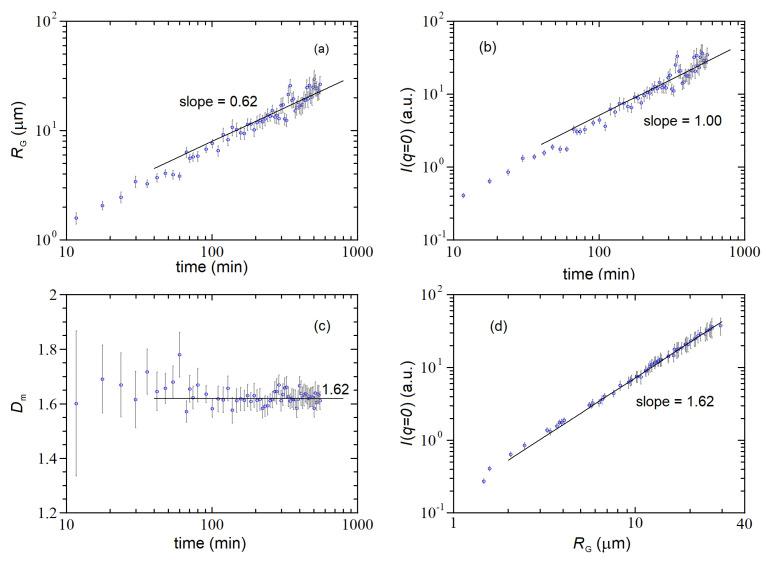
Time behavior of the gyration radius $R_{G}$ (**a**), mass M (**b**) and fractal dimension $D_{m}$ (**c**) for the same aggregation clusters of Figure 4a; (**d**) behavior of zero-q intensity $I\left(q=0\right)$ as a function of the cluster gyration radius $R_{G}$. The three slopes reported in panels (**a**,**b**,**d**) are consistent with a DLCA growth modality characterized by a mass fractal dimension $D_{m}=1.62$.

**Figure 6 nanomaterials-12-02214-f006:**
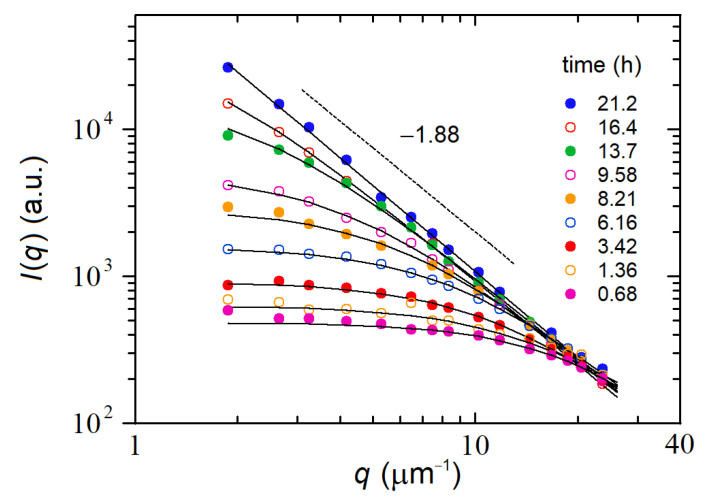
Aggregation of HAPs studied by WA-SLS: behavior of the scattered intensity $I\left(q\right)$ data (symbols) and fits (continuous lines) as a function of q for different times after mixing of the three solutions described in the text. The slope indicates that the clusters have a mass fractal dimension $D_{m}\sim 1.88$.

**Figure 7 nanomaterials-12-02214-f007:**
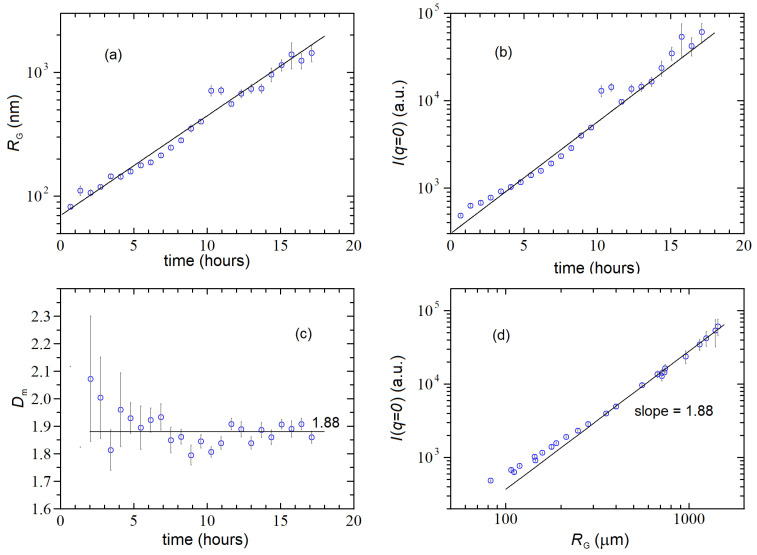
Time behavior of the gyration radius $R_{G}$ (**a**), mass M (**b**) and fractal dimension $D_{m}$ (**c**) for the same aggregation clusters of Figure 6a; (**d**) behavior of zero-q intensity $I\left(q=0\right)$ as a function of the cluster gyration radius $R_{G}$. The three slopes reported in panels (**a**,**b**,**d**) are consistent with an RLCA growth modality characterized by a mass fractal dimension $D_{m}=1.88$.

**Figure 8 nanomaterials-12-02214-f008:**
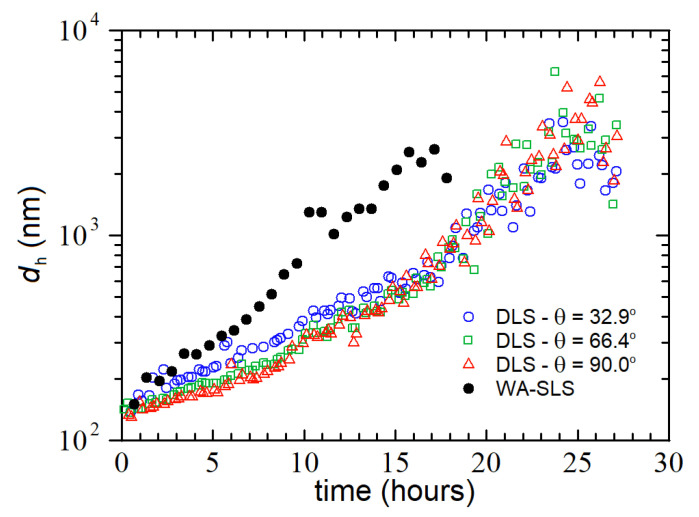
Comparison between the time evolution of the hydrodynamic diameters recovered via DLS at three different angles (open symbols) and the one recovered by using WA-SLS (dots). The latter one was estimated from the $R_{G}$ values (see Figure 7a), as described in the text.

**Figure 9 nanomaterials-12-02214-f009:**
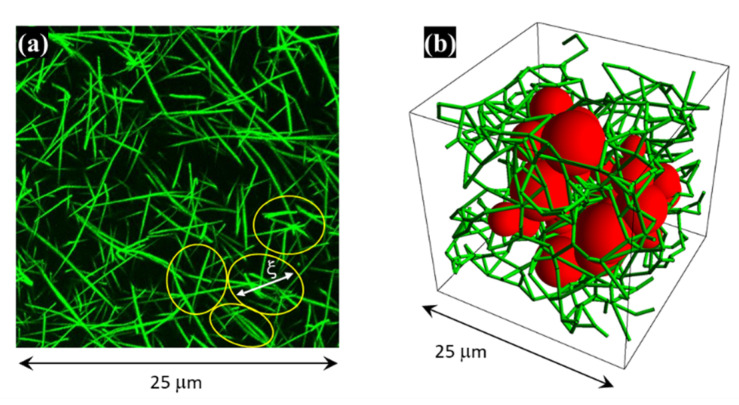
(**a**) Rendering of confocal optical microscope images taken on an aged fibrin gel. The yellow ellipses indicate regions of higher fiber density; (**b**) 3D representation of an (in-silico) fibrin gel, where in red are shown the spheres with the largest diameters that can be optimally fit (tangent to the fibers) in the pore zones of the gel. Their average diameter corresponds to the gel pore size. Pictures adapted with permission from Refs. [41,42].

**Figure 10 nanomaterials-12-02214-f010:**
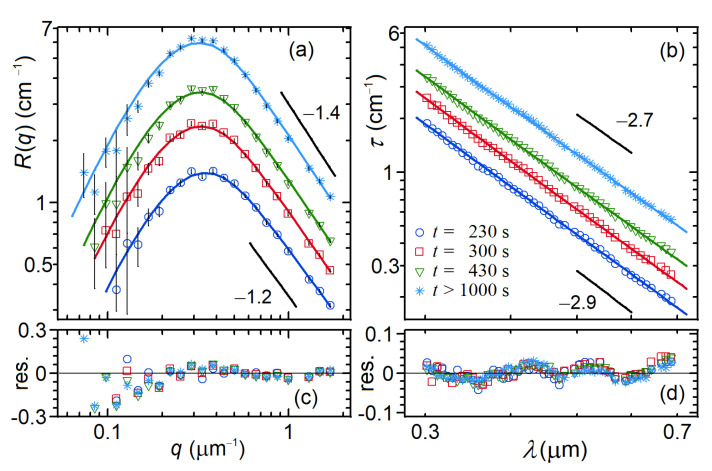
LAELS (**a**) and turbidity (**b**) data taken on a polymerizing fibrin solution (FG 0.45 mg/mL in TBE, THR:FG molar ratio 1÷100) at different, relatively long times after adding thrombin to the FG solution. Continuous lines through the data represent the fittings carried out with the two steps procedure described in the text. Panels (**c**,**d**) report the corresponding relative residuals for each fit. Pictures reproduced with permission from Ref. [14].

## Data Availability

Not applicable.

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
