# Peer review of "Light Scattering and Turbidimetry Techniques for the Characterization of Nanoparticles and Nanostructured Networks"

_nanomaterials, 2022, doi:10.3390/nano12132214_

Round 1

Reviewer 1 Report

The review paper of Anzini et al. combines a concise description of three different light scattering techniques and a presentation of three examples of nano-aggregation studies performed with those techniques by the authors. It is somewhere in between the textbook on the topic and a thorough review of state of the art in the field, which should, in my opinion, include much more examples of similar investigations also performed by other research groups, not only by the authors. Nevertheless, the paper is very well written and gives the reader a useful overview, so I recommend the publication nevertheless.

Regarding theoretical background – I strongly recommend that the authors check the formulas line-by-line, to get rid of the remaining mistakes. In line 158 the refractive index difference should be in the numerator, not the denominator. In line 363, a square is missing in the denominator of the second term, in line 389 d(theta) is missing in the description of the element of spatial angle, in line 861 the quantity A is not defined…. Such errors cause a lot of trouble for newcomers to the field when studying from this type of paper.

As regards the experimental part – the experiments and the results are well explained and nicely presented. The only unclear point that I detected is in Fig 5.c and Fig.7.c where mass M is supposed to be plotted, but the label on a vertical axis in the plot is light intensity at q=0. The authors should either change this or inform the reader where the relationship between I(q=0) and M can be found.

Author Response

We thank the Reviewer for having read so carefully our manuscript and for appreciating the writing. However, his/her main concern regards the fact that only three application examples coming from the authors research group have been reported, with no mention to other investigations performed by different scientists. The Reviewer is right, but this work is not intended to be “a thorough review of state of the art in the field”, which would require a completely different approach and effort.

This article is a mini-review on the LS techniques aimed at summarizing in a concise and useful form the main formulas that characterize these techniques. The reported examples are clearly partial and biased in favour of our past and current research activities, but, exactly for this reason, we had the possibility to enter into the details of the methods and discuss both their potentialities and limitations. Reporting on a such analysis from other scientists’ results would have not been so straightforward and (probably) less insightful.

Anyway, spurred by the Reviewer’s criticism, we added a final paragraph in the conclusions warning the reader about our choice to report only “our” research group examples. The paragraph reads:

         “Finally, we would like to point out that in this mini-review we (deliberately) reported only some examples taken from our research group activity. We are perfectly aware that these examples are quite limited and they are not at all fully representative of the wide spectrum of experimental implementations and applications of the LS techniques. However, thanks to them, we had the possibility of entering the details of the various methods and discussing both their potentialities and limitations. Similarly, when presenting the advantages of coupling different methods, we limited ourselves to only LS techniques, without discussing the possibility of combining LS with SAXS and SANS techniques, but only mentioning an application [44]. This example already highlights that the combined use of LS and SAXS/SANS techniques would clearly enlarge the types of samples (and the range) investigable with scattering techniques, but this was beyond the purpose of this article.”

Regarding theoretical background, we checked again all formulas and corrected the errors and missing definitions pointed out by the Reviewer at lines 162, 368, 394 and 885 (see highlighted revised manuscript).

As to the unclear points of the experimental part relative to Figs. 5c and 7c, the relationship between  and  was recalled at line 547, where we reported the sentence; “[note that  is proportional to , see Eq.(22)]”.

Please, note that we renumbered all citations starting from Ref.[18], which was for some reason missing (probably due to a Zotero error). Therefore, from number 18, the new corrected citation numbers are equal to the old ones + 1.

Reviewer 2 Report

1. In Abstract, at lines 22-24 you mention about the opportunity to combine together different scattering techniques. Then, in Conclusion you mention about LA-SLS, WA-SLS, MWT and DLS. It is not clear why you limited your discussions only to these methods, and omitted other well-established complementary scattering techniques such as SAXS/ SANS, and which extend SALS both in the type of samples suitable for measurements and in q-range. Yes, you mention about SAXS at line 768 but very briefly and in a limited context. I think the manuscript would be more interesting if SALS is related to SAXS/SANS and their similarities and differences are clearly underlined. This can be done probably in 1-2 paragraphs in Introduction section. However, if you plan to include also some SAXS/SANS experimental data, then additional details should be provided in Theoretical background section.

2. In Fig. 10a you provide a graph of experimental R(q) vs. q together with a fit. While the parameters used are indicated at lines 737-738, it is not clear what is the exact expression for R(q)? Presumably this is given by Eq. (13) in Ref. [14] but I think it shall be indicated in the manuscript. Maybe I miss something, but is not clear how R(q) (either given by Eq.(13) in Ref. [14] or not) describes the increasing data at low-q values (up to the peak at about 0.22 inverse micrometers).

3. At lines 566-567 you mention that the exponent -4 expected from Rayleigh scattering is modified into an effective exponent of -4.4. Why exactly -4.4 and not other values?

Round 2

Reviewer 2 Report

The Authors took into account my remarks.